# Revealing firn structure at Dome A region in East Antarctica using cultural seismic noise

Zhengyi Song<sup>1,2</sup>, Yudi Pan<sup>1,2\*</sup>, Jiangtao Li<sup>1,2</sup>, Hongrui Peng<sup>2,7</sup>, Yiming Wang<sup>1,2</sup>, Yuande Yang<sup>3,4</sup>, Kai Lu<sup>5</sup>, Xueyuan Tang<sup>5,6</sup>, and Xiaohong Zhang<sup>3,4,2\*</sup>

- 5 School of Earth and Space Science and Technology, Wuhan University, Wuhan, 430079, China.
  - <sup>2</sup>School of Geodesy and Geomatics, Wuhan University, Wuhan, 430079, China.
  - <sup>3</sup>Chinese Antarctic Center of Surveying and Mapping, Wuhan University, Wuhan, 430079, China.
  - <sup>4</sup>Key Laboratory of Polar Environment Monitoring and Public Governance, Ministry of Education, Wuhan University, Wuhan, 430079, China.
- 5Key Laboratory of Polar Science, Ministry of Natural Resources, Polar Research Institute of China, Shanghai, 200136, China.
  6School of Oceanography, Shanghai Jiao Tong University, Shanghai, 200230, China.
  - <sup>7</sup>Department of Earth and Planetary Sciences, Jackson School of Geosciences, University of Texas at Austin, Austin, TX 78712, USA.

Correspondence to: Yudi Pan (yudipan@whu.edu.cn) and Xiaohong Zhang (xhzhang@sgg.whu.edu.cn)

Abstract. Antarctica is mostly covered by snow, firn, and glacier ice, and the transformation from snow to firn and glacier ice influences energy transfer and material transport in polar regions. In this paper, we deployed three linear seismic arrays near Dome A in East Antarctica during China's 39th and 40th Antarctic scientific expeditions and used seismic ambient-noise to reconstruct the firn structure nearby. The result shows that the ambient-noise mainly comes from the Kunlun Station and is related to human activities. We resolved the empirical Green's function that contains abundant multi-modal surface waves from 3 to 35 Hz, and reconstructed the shallow S-wave velocity, density, and radial anisotropy structures by inverting them. The reliability of the structure was validated by the ice-core data, which demonstrates the effectiveness of using cultural seismic noise for the reconstruction of shallow structures in Antarctica. The result shows that the S-wave velocity increases rapidly with a weak negative radial anisotropy (SH wave travels slower than SV wave) in the top 28 m, which corresponds to the transformation from snow to firn. The firn layer shows a fairly strong positive radial anisotropy (SH wave travels faster than SV wave) between 40 m and 70 m in depth. The radial anisotropy vanishes to zero at around 84 m in depth, denoting the transformation from firn to glacier ice. Overall, the multi-parameter results clearly show the transformation from snow to ice, and the internal evolution of firn at Dome A region. Furthermore, we compared several existing S-wave velocity profiles of firm structures in West and East Antarctica, which indicates relatively higher S-wave velocities in the shallow regions of West Antarctica.

#### 30 1 Introduction

The majority of Antarctica is covered by glacier ice all year round, holding enough water to raise the global sea level by 58 meters (Fretwell et al., 2013). Antarctica plays a significant role in climate change, sea-level change, and ecosystem evolution.

Firn, formed through snow accumulation and subsequent compaction, represents the transitional layer between snow and glacial ice. It is an essential component of the ice sheet and plays a crucial role in material transport (MacAyeal, 2018). Its structure and evolution are influenced by processes such as densification, settling, and refreezing, which are highly sensitive to temperature variation, surface accumulation, and wind patterns (Ligtenberg et al., 2011; Wilkinson, 1988). Understanding firm dynamics is essential for accurately assessing surface mass balance, especially in Antarctica and Greenland (Gardner et al., 2018; Kowalewski et al., 2021; Velicogna et al., 2020). Moreover, firn modulates the depth at which atmospheric gases are sealed into the ice, directly impacting the interpretation of ice core records and paleoclimate reconstructions (Schwander et al., 1997). Variations in firn density and related physical properties affect the retrieval and interpretation of ice sheet elevation changes (Medley et al., 2022; Smith et al., 2023). Firn layers can store meltwater seasonally in the form of firn aquifers, influencing subglacial hydrology and potentially enhancing basal sliding (Forster et al., 2014; Miller et al., 2018). These multifaceted roles establish firm as a critical component in both observational and modeling efforts aimed at improving our understanding of polar ice sheet evolution and mass changes. The current measurement of ice sheet mass changes often relies on satellite altimetry (Boening et al., 2012). Furthermore, it has low sensitivity to the firn layer, making it difficult to distinguish whether changes in the ice sheet are due to surface accumulation (increased snowfall) or internal density changes (firn compaction; Ligtenberg et al., 2011; Yang et al., 2024). Consequently, complementary approaches are essential to provide a more detailed characterization of the subsurface structure, thereby enabling a more precise assessment of ice sheet changes. Seismic ambient-noise interferometry has been widely used to study subsurface structure (Nakata et al., 2019). Ambient noises related to both natural and cultural sources are used to characterize the Earth models across scales (Mi et al., 2022; Liu et al., 2023; Zhao et al., 2023; Feng et al., 2024). Seismic interferometry utilizes the cross-correlation functions (CCFs) of the seismic wavefields between station pairs to obtain the empirical Green's function, which can be used to analyze the S-wave velocity and anisotropic structure. Ambient-noise tomography has been successfully used in Antarctica to obtain its structures down to hundreds to thousands of meters (e.g., Fu et al., 2022; Chaput et al., 2022). Similar ambient-noise studies have also been performed in Greenland, Glacier d'Argentière (France), Gornergletscher (Switzerland), Aletschgletscher (Switzerland) and de la Plaine Morte (Switzerland) (e.g., Pearce et al., 2024; Sergeant et al., 2020; Preiswerk and Walter, 2018; van Ginkel et al., 2025) to investigate glacier structures. These studies provided important insights into the subsurface structure of firn and ice. For the whole Antarctica, most of the existing studies on the firn structure are in the WAIS, mainly because more scientific activities were conducted in West Antarctica. However, in recent years, seismic noise methods along with seismic activesource, gravity, radar, and ice core measurements (e.g., Qin et al., 2024; Yang et al., 2024; Zhang et al., 2022; Yang and Li, 2022; Tang et al., 2020) have played an increasingly important role in advancing scientific exploration in the EAIS. Dome A is the highest point of the EAIS with an elevation of about 4093 m. It maintains extreme environmental conditions characterized by low temperatures and low snow accumulation rates (Hou et al., 2007). It preserves ice core records with extended temporal continuity and stratigraphic integrity, maintaining paleoclimate information that enhances our understanding of global climate change (Ma et al., 2010). Some studies suggest that Dome A may contain ice older than 1 million years (Sun et al., 2014). Although the shallow firn layer is known to be important, detailed investigations on the East

Antarctic Plateau are still limited. Prior studies have demonstrated that the mechanical properties of firn can be influenced by ice crystal anisotropy at depths down to 100 m (Schlegel et al., 2019; Gerber et al., 2023; Pearce et al., 2024). Thus, applying seismic ambient-noise studies at Dome A can provide new insights into firn structure across high-elevation regions of the East Antarctic Plateau. Furthermore, investigations of its meteorological conditions, surface geomorphology, ice thickness, subglacial topography, ice flow, and internal stratigraphy offer key constraints on the evolution of EAIS (Tang et al., 2012). In this study, we collected three-component ambient-noise data during China's 39th and 40th Antarctic scientific expeditions to study the shallow structure at Dome A region in the EAIS. After processing and analyzing the data, we obtained the multi-modal dispersion curves of Rayleigh- and Love-waves and derived the SV- and SH-wave velocity ( $V_{SV}$  and  $V_{SH}$ ), respectively. The presence of human activities, acting as stable seismic sources, improves the robustness of our results. We used the data to investigate the firn structure at Dome A region by reconstructing its S-wave velocity, density, and radial anisotropy profiles. These results are compared with previous studies in other regions in Antarctica to analyze the shallow firn structures.

## 2 Data and Methods

We deployed a total of 73 three-component seismic nodes near Dome A and the Kunlun Station, which is located about 7.3 km southwest of Dome A (Fig. 1a). In this study, we mainly used 19 seismic nodes with a 500 m interval (Line 1) to reconstruct the shallow structure at Dome A region. In addition, 18 seismic nodes deployed with a 500 m interval on the south side of the Kunlun station (Line 2), and 7 seismic nodes deployed on the north part of Line 1 with irregular intervals ranging from  $\sim$ 90 m to 1000 m (Line 3) were used to analyze the distribution of ambient-noise source and to help identifying multi-modal dispersion curves, respectively. Three lines acquired the seismic data with a length of 9 days, 6 days, and 14 days, respectively. Line 1 and Line 2 are located to the northeast and southeast of the Kunlun Station, respectively. Dome A is located approximately at the central point of Line 1. Unlike the study which shows that inland seismic noise in Antarctica is typically dominated by the secondary and primary microseism bands (Anthony et al., 2015), the dominant frequencies of the data are about 8 Hz and 15 Hz, indicating strong high-frequency noise in this region. We analyzed the ambient-noise source of seismic data in Line 1 and Line 2 by beamforming. Figures 1b-e show the energy distribution of ambient-noise for the two linear arrays at the two dominant frequencies, respectively. The energy peaks in Line 1 (Fig. 1b and c) are more prominent than those in Line 2 (Fig. 1d and e), suggesting a more concentrated noise source in the data acquired by Line 1. Results of Line 1 and Line 2 show a unidirectional noise source at both 8 Hz and 15 Hz. The dominant source is around 217° in the data acquired by Line 1 and 340° in Line 2. It suggests that the ambient-noise source mainly comes from the Kunlun Station (Fig. 1a) and is mainly related to human activities during China's 39th Antarctic scientific expedition. The primary peak at 8 Hz corresponds to the fundamental-mode surface wave, while the two peaks at 15 Hz correspond to higher modes. In other words, we mainly used cultural ambient-noise in this paper to reconstruct the near-surface structure at Dome A region.

**Figure 1.** Arrays information and estimation of ambient-noise source location. (a) Line 1 (black triangles) and Line 2 (black hollow triangles) were collected in January 2023, and Line 3 (white triangles) was collected in January 2024. The Kunlun Station (red pentagram), Dome A and the ice core (DA2005) (red circle) are also marked. The inset provides an overview of the study location (green circle) on the whole Antarctica. Beamforming results of the vertical component at frequencies (b) 8 Hz and (c) 15 Hz for Line 1, and (d) and (e) at the same frequencies for Line 2. The back azimuths are from 0° (north) to 360° and slowness is from 0.4 to 1.0 s km<sup>-1</sup>. Two arrows in (a) denote the primary back azimuths from the beamforming results.

We processed the ambient-noise data acquired by Line 1 and Line 3 using seismic interferometry (Bensen et al., 2007). We first cut the ambient-noise data into 10-minute segments. Then, we applied both running absolute mean normalization and spectral whitening to the data. Subsequently, we used cross-correlation and phase-weighted stacking (Schimmel and Paulssen, 1997) to recover empirical Green's functions. The obtained CCFs of vertical-vertical (ZZ), radial-radial (RR), and transverse-transverse (TT) components are strongly asymmetric (Fig. 2a and c) due to the existence of a strong directional source. The CCFs that correspond to the cultural seismic noise from the Kunlun Station are clear and of high signal-to-noise ratio (up to 80 dB).

Based on the CCFs, we obtained the dispersion spectra of Line 1 and Line 3 (Fig. 2b and d) by using the phase shift method (Park et al., 1999), respectively. Relatively strong spatial aliasing (Cheng et al., 2023), which is caused by relatively large inter-station distances (~500 m), is observed in the high-frequency range of data in Line 1 (Fig. 2b). Line 3 was designed with irregular spacing containing a shortest inter-station distance of 90 m, which greatly helps with the identification of multi-modal surface-wave dispersion curves. By considering the dispersion spectra from CCFs of Line 1 and Line 3 simultaneously, we can obtain multi-modal dispersion curves accurately. A total of five modes in Rayleigh waves and four modes in Love waves

are identified (arrows in Fig. 2b and d), while the first three modes are recognized with relatively higher confidence (white arrows in Fig. 2b and d). Noise data from Line 1 and Line 3, which were collected at different times, show high consistency in the multi-modal dispersion curves. The estimated dispersion curves agree with the slowness values in the beamforming results (Fig. 1b and c), which further proves the reliability of the multi-modal dispersion curves.

**Figure 2.** Ambient-noise CCFs and dispersion spectra. (a) CCFs of vertical-vertical (ZZ), radial-radial (RR), and transverse-transverse (TT) components between 3 and 35 Hz for Line 1. (b) shows the dispersion spectra corresponding to the CCFs in (a). (c) and (d) CCFs and dispersion spectra for Line 3. The white arrows with numbers 0, 1, 2 in (b) and (d) show the picked fundamental-mode, first high-mode, and second high-mode dispersion curves. The black arrows indicate potential higher modes that are not used in the inversion.

#### 3 Inversion and Results

We invert the dispersion curves of the first three modes simultaneously with a multi-objective function (Wang et al., 2023):

$$\boldsymbol{\varphi}(m) = \left[ \left\| d_0^{syn} - d_0^{obs} \right\|, \left\| d_1^{syn} - d_1^{obs} \right\|, \left\| d_2^{syn} - d_2^{obs} \right\| \right], \tag{1}$$

in which  $d_0$ ,  $d_1$ , and  $d_2$  represent the dispersion curve of the fundamental mode, first high mode, and second high mode, respectively;  $d^{syn}$  and  $d^{obs}$  represent the synthetic and observed dispersion curves, respectively. The forward simulation of dispersion curves is solved with Knopoff's method (Knopoff, 1964). For each frequency, we can obtain a set of solutions that correspond to fundamental and higher modes sequentially. In this algorithm, we treat the fitting of dispersion curves corresponding to different modes independently in a vector-valued multi-objective function (Eq. 1).

The multi-objective optimization problem is solved with a multi-objective particle swarm optimization algorithm (MOPSO).

The inversion result consists of a group of Pareto optimal solutions, which represent the best solutions in a multi-objective optimization problem. For a Pareto solution, we cannot decrease one of its objective values without increasing at least one of the other two objective values (Eq. 1). The difference among the Pareto optimal solutions represents the trade-off among the

three objectives.

145

150

The  $V_{SH}$  profile is estimated by using the TT component, and  $V_{SV}$  profile is resolved by using ZZ and RR components simultaneously. A 20-layer model with its S-wave velocity varying between 300 and 2100 m s<sup>-1</sup> is used as the search space. Similarly, the P-wave velocity is limited between 600 and 4100 m s<sup>-1</sup>. The density is evaluated from  $V_{SH}$  using the empirical relationship presented in Diez et al. (2014):

$$\rho(z) = \frac{\rho_{ice}}{1 + [(V_{S,ice} - V_S(z))/950]^{1.17}},$$
(2)

where  $\rho_{ice}$  and  $V_{S,ice}$  are the density and S-wave velocity of ice, respectively. We set a value of 900 kg m<sup>-3</sup> for  $\rho_{ice}$  from Zhang et al. (2022) and 1960 m s<sup>-1</sup> for  $V_{S,ice}$ , which is estimated from the phase velocity of higher modes at their cut-off frequencies (Fig. 3).

Taking the Love wave as an example, the inversion converges after 300 iterations and provides 83 Pareto optimal solutions. Similarly, we obtained 78  $V_{SV}$  models from Rayleigh-wave case. In both  $V_{SH}$  and  $V_{SV}$  results, the Pareto solutions show fairly high consistency (Fig. 3a and b), especially in the region between 30 and 70 m in depth. The knee point solution, which is located at the convex region on the surface defined by the Pareto solution set, shows high similarity with the mean value of the Pareto solutions (red and blue curves in Fig. 3a and b). The fitting between the multi-modal dispersion curve is fairly good (Fig. 3d and e), not only for the first three modes used in the inversion, but also for the fourth and fifth modes that are not used in the inversion (modes with arrows in Fig. 3d and e).

Figure 3. Inverted firn structure and model fitting. (a) 1-D  $V_{SH}$  profile estimated by using the TT component and (b) 1-D  $V_{SV}$  profile estimated by using ZZ and RR components from Line 1. The grey, red, and blue lines represent all Pareto optimal solutions, the knee point solution, and the mean solution, respectively. (c) Density estimated by empirical relationship given in Eq. (2) with  $V_{SH}$ , and the filled circles show the ice core (DA2005) density near Dome A. Fitting between the observed and synthetic dispersion curves for (d) Love- and (e) Rayleigh-wave. The black arrows denote the modes that have not been used in the inversion.

We compare the density profiles corresponding to the *V<sub>SH</sub>* results with ice-core data (DA2005) collected around 300 m away from Dome A (Jiang et al., 2012; Yang et al., 2021). The core density was measured in situ using the weight-volume method (Xiao et al., 2008). The RMSEs between the density (mean and knee point results) and ice core data are less than 44 kg m<sup>-3</sup> (Table A1 in Appendix A), and our model only shows some deviation from the ice-core data in the shallow part (depth < 20 m). Three possible reasons for the difference are the errors in the velocity-density empirical relationship for the shallow air-snow layer, the temporal (18 years between the times when the ice-core and seismic data were acquired) and spatial variation in the glacier (the ice core is 500 m away from our survey line), and the relatively low sensitivity of our observed dispersion

curve due to the lack of short-wavelength data. The RMSEs between the synthetic and observed dispersion curves are less than 34 m s<sup>-1</sup> (Table A2 in Appendix A). Overall, the relatively low RMSE values suggest that our reconstructed models are reliable.

## 4 Shallow structure at Dome A region

- The inversion results show a notable discrepancy between  $V_{SH}$  and  $V_{SV}$  (Fig. 3a and b), which indicates the presence of radial anisotropy in the firn layer (Fig. 4a). For each solution in the  $V_{SH}$  Pareto set, we calculated the differences with every solution in the  $V_{SV}$  Pareto set (grey lines in Fig. 4a). All solution-to-solution anisotropy results and the mean value (red line in Fig. 4a) show a clear depth-dependent trend. The results show that  $V_{SV}$  is faster than  $V_{SH}$  in the shallow part, while their relationship is reversed after reaching the first critical depth (28.1 m, corresponding to the critical density of 550 kg m<sup>-3</sup>).
- Taking into consideration of both density and anisotropy profiles, we can resolve the variation of shallow material with depth at Dome A region. The material transforms from snow to firn and eventually to glacier ice within the top 100 m at Dome A region (Fig. 4b). The shallow material contains freshly fallen snow, which typically takes a hexagonal shape and is filled with air. With the rounding of snow grains and the densification process (Amory et al., 2024), the snow becomes more spherical. Accompanied by the settling and grain growth, snow gradually transforms into firn, and the process continues as the density reaches 400 kg m<sup>-3</sup>. It leads to a fast increase in density and the disappearance of the weak radial anisotropy when the first critical depth (~ 28 m) is reached.
- Subsequently, as the depth continues to increase, the densification rate decreases, and the recrystallization and deformation processes become dominant. The radial anisotropy gradually increases with depth, and reaches a maximum of approximately 11 %. As the density increases to 800 kg m<sup>-3</sup>, sintering occurs in firn, and radial anisotropy decreases accordingly. Reaching the second critical depth at 84.3 m, which corresponds to the pore close-off density (830 kg m<sup>-3</sup>), the firn gradually transforms into glacier ice, and radial anisotropy decreases to approximately zero.

Figure 4. Radial anisotropy and shallow material transformation. (a) Radial anisotropy  $(V_{SH}/V_{SV} - 1)$  (grey lines) for all pairs of  $V_{SH}$  and  $V_{SV}$ , along with the mean (red line) solution. Mean density (blue line) from Fig. 3c is also denoted. (b) Schematic picture of possible shallow structure in our study area.

190

200

Previous studies (e.g., Pearce et al., 2024; Chaput et al., 2023; Diez et al., 2016; Picotti et al., 2015; Schlegel et al., 2019) show that firn anisotropy may be caused by three primary mechanisms: (1) effective anisotropy, related to very thin layers formed during firn densification; (2) structural anisotropy, related to fractures or microcracks caused by non-isotropic stress; and (3) intrinsic anisotropy, associated with preferred crystal orientation.

In the shallow layer (< 20 m), some lateral variation may exist along Line 1 (~10 km). Our observations lack data at shorter wavelength ranges (< 40 m), and the weak anisotropy results (< 5 %) in the shallow layer exhibit high uncertainty (Fig. 4a). Therefore, we refrain from further interpretation of the anisotropy within the upper 20 m.

The mean accumulation rate in the Dome A region is about 2.3 cm water equivalent per year (Jiang et al., 2012) and, therefore, firn densification can produce millimeter-scale layers whose thicknesses are much smaller than the seismic wavelengths observed in our study. These thin layers lead to different elastic properties in the vertical and horizontal directions (Diez et al., 2016; Schlegel et al., 2019). It makes the SH waves travel relatively faster than SV waves along the horizontal direction, which is consistent with our observation.

The average strain rate in the Dome A region ( $\sim 1.6 \times 10^{-12} \text{ s}^{-1}$ ) (Yang et al., 2014) is much lower than the typical ductile to brittle transition values for firn ( $\sim 10^{-4}$  to  $10^{-2}$  s<sup>-1</sup>) (Narita, 1984; Kirchner et al., 2001), suggesting that tensile stresses are

unlikely to cause brittle crevasse in our study area. Therefore, the structural anisotropy caused by crevasse is not considered the main reason for anisotropy in the Dome A region.

Regarding intrinsic anisotropy, the preferred orientation direction of ice crystals is related to gravitational compaction and ice flow. For the firn layer after reaching the first critical depth (28.1 m), slow ice flow (~11.1 cm a<sup>-1</sup>) that is perpendicular to the Line 1 (Yang et al., 2014) causes re-orientation of the crystal along the horizontal direction. This leads to SH waves traveling faster than SV waves. Overall, we infer that the observed anisotropy at Dome A region is primarily attributed to effective anisotropy and intrinsic anisotropy.

## **5 Discussion**

In our study, the cultural seismic noise provides an efficient source for the accurate reconstruction of shallow velocity structures. The surface waves remain clear in the CCFs in the station 12 km away from the Kunlun Station (last trace in Fig. 2a), thanks to the dominance of energetic cultural noise and the simple horizontally layered structure with a low level of scattering effect. It shows that the noise generated by daily activities or research activities at scientific camp can rapidly and accurately reveal the shallow structure within tens of kilometers, which significantly reduces workload compared with explosive seismic source.

We also compare the 1D near-surface S-wave velocity profiles acquired from different regions of Antarctica (Fig. 5a and b), including Dome A (this study), the Ross Ice Shelf (Diez et al., 2016), the Rutford Ice Stream (Zhou et al., 2022), the Whillans Ice Stream (Picotti et al., 2024), the WAIS Divide camp (Qin et al., 2024; Zhang et al., 2024), and the Polar point (Yang et al., 2024). All of them show a rapid increase in S-wave velocity within the upper 20 to 30 m, followed by a gradual decrease in the rate of increase, ultimately approaching the ice velocity of around 2000 m s<sup>-1</sup> (Fig. 5a).

Figure 5a suggests relatively higher  $V_S$  in the shallow regions in West Antarctica. However, it should be noted that the firm densification process exhibits regional variability and is influenced by multiple factors, such as temperature and ice velocity. The annual mean temperature in the West Antarctic region is generally higher than that of the East Antarctic Plateau by approximately 20–30 °C (Nielsen et al., 2023; Wang et al., 2023), which may contribute to the differences in velocity structure. Different localized ice velocities, heat flux, sublimation, and blowing snow erosion (Goujon et al., 2003) at these specific sites might also influence the shallow structure. Although Fig. 5a indicates a notable difference in the  $V_S$  structures between East and West Antarctica, the observations are not sufficiently large to draw conclusion at the current stage.

In addition, Line 1 and Line 2 are oriented in different azimuths, making the investigation of azimuthal anisotropy possible. The azimuthal anisotropy might provide additional information about the direction of ice flow or the orientation of crystal fabric, and strengthen our knowledge about the ice in the inland area of East Antarctica. The estimation of azimuthal anisotropy deserves further study in the future.

Figure 5. S-wave velocity from East and West Antarctica. (a) East Antarctica:  $V_{SH}$ ,  $V_{SV}$  from this study (blue line with pentagrams and blue line), and  $V_{SV}$  from Yang et al. (2024) (green curve with circles). West Antarctica:  $V_{SH}$  and  $V_{SV}$  from Diez et al. (2016) (pink curve with triangles and pink dotted curve with triangles),  $V_{SV}$  from Zhou et al. (2022) (yellow dotted curve with diamonds),  $V_{SH}$  from Picotti et al. (2024) (red dash-dot curve with inverted triangles),  $V_{SV}$  from Qin et al. (2024) (brown dashed curve with squares), and  $V_{SV}$  from Zhang et al. (2024) (purple line with squares). The markers in (b) corresponding to (a) represent the respective study areas from different research works in Antarctica.

## **6 Conclusion**

We analyzed the seismic ambient-noise data and reconstructed shallow subsurface models at Dome A region in Antarctica. The results of beamforming showed that the seismic ambient-noise source mainly comes from the Kunlun Station and is related to human expedition works. We obtained multi-modal surface-wave dispersion curves up to 35 Hz using seismic interferometry and reconstructed both  $V_{SH}$  and  $V_{SV}$  models down to about 100 m depth. The inversion results, including both the density profile derived empirically from the S-wave velocity and the anisotropy profile, show a clear transformation of material from snow to firn and firn to ice, and the internal evolution of firn in the top 100 m. Our density profile nicely agrees with ice-core data nearby, indicating fairly high reliability of the subsurface model reconstructed at Dome A region. Our study also suggests that cultural ambient-noise can serve as a nice source for the investigation of shallow structures in the polar regions.

## Appendix A

255

In Appendix A, we showed the estimated density using different empirical relationships and ice density values, and compared them with ice core (DA2005) density (Fig. A1). We also showed the sensitivity kernels for multi-modal Love and Rayleigh waves (Fig. A2). The  $V_{SH}$  and  $V_{SV}$  solution sets (Fig. 3a and b) are plotted together (Fig. A3) to show their relative relationship. We calculated RMSE between estimated density and ice core (DA2005) density (Table A1) and RMSE of multi-mode dispersion curve fitting (Table A2).

Figure A1. Density estimated by empirical relationship. (a) Using a relationship from Diez et al. (2014) with  $V_{SH}$ . The red and blue lines use an ice density of 900 kg m<sup>-3</sup>, while the green and yellow lines use 920 kg m<sup>-3</sup>. (b) Using the relationship from Yang, Zhan et al. (2024) with  $V_{SV}$  (grey, green and yellow lines). The filled circles show the ice core (DA2005) density near Dome A. The red and blue lines in (a) and (b) are consistent with Fig. 3c in the main text.

**Figure A2**. Depth sensitivity kernels at different frequencies for Rayleigh (a-c) and Love (d-f) waves for the fundamental (mode 0), first high (mode 1), and second high (mode 2) mode. The black dotted lines represent the dispersion curves for each mode.

**Figure A3**. Pareto solution sets of  $V_{SH}$  (light red lines) and  $V_{SV}$  (light blue lines). The mean solutions of  $V_{SH}$  and  $V_{SV}$  are marked by red and blue lines, respectively. These results are the same as those presented in Figure 3a and b of the main text.

Table A1. Root-mean-square errors (RMSE) between estimated density results (mean and knee point results) and ice core (DA2005) density.

| Empirical Relationships                    | $ ho_{ice}$ (kg m <sup>-3</sup> ) | Mean<br>RMSE (kg m <sup>-3</sup> ) | Knee point<br>RMSE (kg m <sup>-3</sup> ) |
|--------------------------------------------|-----------------------------------|------------------------------------|------------------------------------------|
| Diez et al., 2014<br>(SH velocity-density) | 900                               | 43.8465*                           | 43.5244*                                 |
| Diez et al., 2014<br>(SH velocity-density) | 920                               | 46.2571                            | 44.8456                                  |
| Yang et al., 2024<br>(SV velocity-density) | 920                               | 47.0541                            | 42.3032                                  |

<sup>\*</sup> indicates the empirical relationship and ice density value used in the main text.

Table A2. Root-mean-square errors (RMSE) of multi-mode dispersion curve fitting.

| Dispersion Modes     | Love Wave                         |                                         | Rayleigh Wave                     |                                         |
|----------------------|-----------------------------------|-----------------------------------------|-----------------------------------|-----------------------------------------|
|                      | Mean<br>RMSE (m s <sup>-1</sup> ) | Knee point<br>RMSE (m s <sup>-1</sup> ) | Mean<br>RMSE (m s <sup>-1</sup> ) | Knee point<br>RMSE (m s <sup>-1</sup> ) |
| Fundamental (Mode 0) | 22.3250                           | 17.0714                                 | 33.9394                           | 25.7128                                 |
| First high (Mode 1)  | 28.0159                           | 28.1090                                 | 22.5358                           | 29.8653                                 |
| Second high (Mode 2) | 11.9490                           | 8.3569                                  | 18.2726                           | 19.9725                                 |
| Third high (Mode 3)  | 14.0678*                          | 15.2696*                                | 14.5797*                          | 14.3996*                                |
| Fourth high (Mode 4) | _                                 | _                                       | 21.4708*                          | 28.0765*                                |

<sup>\*</sup> indicates the mode not involved in the inversion of dispersion curve.

# Data availability

The density,  $V_{SH}$  and  $V_{SV}$  models in this study are available at Song et al. (2025).

## **Author contribution**

YP, JL, YY, KL, XT and XZ planned the campaign; YY and KL performed the measurements; ZS, YP, JL, HP and YW analyzed the data; ZS and YP wrote the manuscript draft; JL, HP, YY, KL, XT and XZ reviewed and edited the manuscript.

## **Competing interests**

The authors declare that they have no conflict of interest.

## Acknowledgments

This study is funded by the National Natural Science Foundation of China (Grant No. 42474190, 42174069, 42276257) and the Fundamental Research Funds for the Central Universities (Grant No. 2042025kf0077). We thank the editor Dr. Adam Booth, Dr. Yan Yang and two other anonymous reviewers for their constructive comments.

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
