# Peer review of "Revealing firn structure at Dome A region in East Antarctica using cultural seismic noise"

_EGUsphere, 2025_

## Author Comment (AC1)

**Response to Reviewer 1 (Yan Yang)**

**General comments:**

This manuscript presents results using high-frequency cultural seismic noise from the Kunlun Station to image the shallow firn structure in the Dome A region of East Antarctica. The work resolves S-wave velocity and radial anisotropy down to ~100 m in the firn and validates the results with nearby ice-core data and results from other sites in Antarctica. The study offers an application of passive seismic methods in a remote polar region with limited prior coverage. The paper is well organized. The results are well illustrated. The implications for regional differences in firn compaction and accumulation rates are relevant. Overall, I believe this manuscript is well suited for publication in The Cryosphere after minor revisions.

**Dear Dr. Yan Yang,**

Thank you very much for your constructive comments and suggestions! We have carefully revised our manuscript accordingly. Please find our point-to-point responses below.

The line number used in this response letter refers to the manuscript with marks.

**Specific comments:**

1. In Figure 3c, the density model generally agrees with borehole studies, but some misfit is still present—specifically, overestimation below ~50 m and underestimation above ~30 m. A similar misfit pattern is reported in the cited study by Yang, Zhan et al. (2024), which motivated the development of an East Antarctica–specific empirical velocity–density relationship to better match observed firn density profiles. I see that you use Equation (2) from Diez et al. (2014), which is based on SH-wave velocity. Since you also resolve Vsv and your site is in the East Antarctic Plateau, I am curious how the results would compare if you applied the Yang, Zhan et al. (2024) relationship using your Vsv model. Additionally, Equation (2) assumes an ice density of 900 kg/m³—would using a more conventional value such as 920 kg/m³ change your results significantly? I understand the need for site-specific relations, but a brief comparison or discussion would strengthen this section.

We followed your suggestion and compared the results obtained using different parameter values in the empirical relationship.

(1) The results using different ice densities (900 and 920 kg m⁻³, Figure A1a) are very similar. Relatively speaking, the estimated density profile using an ice density of 920 kg/m³ is slightly closer to the borehole data (DA2005) in the region shallower than 50 m depth, and slightly further from DA2005 in the region deeper than 50 m depth (blue and yellow lines in Figure A1a). It indicates that the choice of ice density has a minor impact on the result.

(2) In Figure A1b, the density result also generally agrees with the borehole data (DA2005). The result based on the relationship from Yang, Zhan et al. (2024) improves the fit within the upper ~30 m compared to that of Diez et al. (2014) but also shows some deviations at greater depths. The RMSE results (Table A1) indicate that the densities estimated using different empirical relationships and ice density values generally agree with the ice core data.

(3) We also agree that a site-specific velocity-density relationship may be beneficial. Given the spatial distance and difference in acquisition time between the borehole and our linear array, the borehole data serve only as a reference density result. Further borehole measurements,

ideally closer in both space and time to the array, may help to establish a representative empirical relationship near the Dome A region.

We added this information in the Appendix A (Figure A1 and Table A1).

[Figure]

**Figure A1**. Density estimated by empirical relationship. **(a)** Using a relationship from Diez et al. (2014) with $V_{SH}$. The red and blue lines use an ice density of 900 kg m$^{-3}$, while the green and yellow lines use 920 kg m$^{-3}$. **(b)** Using the relationship from Yang, Zhan et al. (2024) with $V_{SV}$ (grey, green, and yellow lines). The filled circles show the ice core (DA2005) density near Dome A. The red and blue lines in (a) and (b) are consistent with Fig. 3c in the main text.

Table A1. Root-mean-square errors (RMSE) between estimated density results (mean and knee point results) and ice core (DA2005) density.

| Empirical Relationships | $\rho_{ice}$ (kg m$^{-3}$) | Mean RMSE (kg m$^{-3}$) | Knee point RMSE (kg m$^{-3}$) |
|---|---|---|---|
| Diez et al., 2014 (SH velocity-density) | 900 | 43.8465[*] | 43.5244[*] |
| Diez et al., 2014 (SH velocity-density) | 920 | 46.2571 | 44.8456 |
| Yang et al., 2024 (SV velocity-density) | 920 | 47.0541 | 42.3032 |

[*] indicates the empirical relationship and ice density value used in the main text.

2. You have cited studies reporting radial anisotropy in firn at levels of 10–15% for several West Antarctic sites. I suggest also citing Schlegel et al. (2019), which examines radial anisotropy at the Kohnen site in East Antarctica. Additionally, I am curious about the robustness of the radial anisotropy inferred above 20 m depth. The cited study Pearce et al. (2024), using similar frequency bands, noted a lack of sensitivity to the top ~20 m in surface-wave inversions and therefore did not interpret their observed shallow radial anisotropy. Could you show the Rayleigh wave sensitivity kernel and comment on whether your inversion results are similarly limited in sensitivity in the

uppermost firn?

We have added an introduction to Schlegel et al. (2019) in the revised manuscript (Lines 77, 212, and 222).
We showed the sensitivity kernels for multi-modal Love and Rayleigh waves (Fig. A2, which is also added in Appendix A in the revised manuscript). The inclusion of higher modes improves the sensitivity of surface waves to the shallow region. However, on the one hand, the sensitivity to the structures shallower than 20 m is overall limited, and on the other hand, the Pareto results (Fig. A3; consistent with Fig. 4a in the original manuscript) show relatively greater uncertainty of radial anisotropy in the shallow region. Therefore, we didn't interpret the result in the top 20 m.

[Figure]

**Figure A2**. Depth sensitivity kernels at different frequencies for Rayleigh (a-c) and Love (d-f) waves for the fundamental (mode 0), first high (mode 1), and second high (mode 2) mode. The black dotted lines represent the dispersion curves for each mode.

[Figure]

**Figure A3**. Radial anisotropy. The figure is taken from Figure 4a of the original manuscript.

3. Lines 1 and 2 are oriented in different azimuths, providing an excellent opportunity to investigate azimuthal anisotropy. Applying the same dispersion analysis workflow to Line 2 could help evaluate directional dependence of seismic velocities, which may relate to ice flow direction or crystal fabric. Is there a reason why dispersion analysis was not performed on Line 2—perhaps due to the absence of a short-spacing array needed for resolving higher modes? Regardless, I suggest including a discussion on the potential for azimuthal anisotropy and how it might be constrained by the existing dataset.

In this manuscript, we mainly used the data along Line 1 in order to estimate the radial anisotropy at the Dome A region. We didn't use the data from Line 2 because it is located further away from Dome A compared to Line 1. We agree that the differing orientations of Line 1 and Line 2 make the investigation of azimuthal anisotropy possible. We added the following content to the "Discussion" in the revised manuscript (Lines 260-263):

"In addition, Line 1 and Line 2 are oriented along different azimuths, making the investigation of azimuthal anisotropy possible. The azimuthal anisotropy might provide additional information about the direction of ice flow or the orientation of crystal fabric, and strengthen our knowledge about the ice in the inland area of East Antarctica. The estimation of azimuthal anisotropy deserves further study in the future."

4. The observed difference in firn density profiles between East and West Antarctica is interpreted as a result of differences in snow accumulation rates. Temperature is another factor that significantly affects firn densification rates. Could you provide information or discussion on the differences in mean annual temperature between your site and the West Antarctic sites included in your comparison?

The comparison between the temperature in East and West Antarctica has been studied by Nielsen et al. (2023) and Wang et al. (2023). Their results show that the annual mean temperature in the West Antarctic region is generally higher than that of the East Antarctic Plateau by approximately 20–30 °C. We added this information in the revised manuscript (Lines 255-256):

"The annual mean temperature in the West Antarctic region is generally higher than that of the East Antarctic Plateau by approximately 20–30 °C (Nielsen et al., 2023; Wang et al., 2023), which may contribute to the differences in velocity structure."

References:
Nielsen, E. B., Katurji, M., Zawar-Reza, P., and Meyer, H.: Antarctic daily mesoscale air temperature dataset derived from MODIS land and ice surface temperature, Sci. Data, 10, 833. https://doi.org/10.1038/s41597-023-02720-z, 2023.
Schlegel, R., Diez, A., Löwe, H., Mayer, C., Lambrecht, A., Freitag, J., Miller, H., Hofstede, C., and Eisen, O.: Comparison of elastic moduli from seismic diving-wave and ice-core microstructure analysis in Antarctic polar firn, Ann. Glaciol., 60, 220-230, https://doi.org/10.1017/aog.2019.10, 2019.

Wang, Y., Zhang, X., Ning, W., Lazzara, M. A., Ding, M., Reijmer, C. H., Smeets, P. C. J. P., Grigioni, P., Heil, P., Thomas, E. R., Mikolajczyk, D., Welhouse, L. J., Keller, L. M., Zhai, Z., Sun, Y., and Hou, S.: The AntAWS dataset: a compilation of Antarctic automatic weather station observations, Earth Syst. Sci. Data, 15, 411–429, https://doi.org/10.5194/essd-15-411-2023, 2023.

---

## Author Comment (AC2)

**Response to Anonymous Reviewer 2**

This paper presents a shear wave velocity model of the firn at Dome A, using established methods previously applied in other parts of the cryosphere. The results—particularly the identification of a relationship between firn compaction and radial anisotropy—are consistent with findings from studies conducted in other regions (e.g., Pearce et al., 2024; Diez et al., 2016). The value of this work lies in demonstrating that these results can be reproduced at Dome A, thereby contributing to our understanding of the consistency and geographic variability of firn anisotropy.

However, I find that the manuscript does not sufficiently establish the glaciological significance of these findings. The broader comparisons made between East and West Antarctica oversimplify the complex and highly localised nature of firn properties. Firn structure and compaction vary significantly across sites, and the paper's attempt to draw sweeping conclusions at the continental scale lacks the nuance required to be scientifically robust.

Moreover, the discussion does not adequately engage with existing literature on firn modelling and radial anisotropy. Key issues, such as intrinsic versus extrinsic causes of anisotropy, have been addressed more thoroughly in other studies yet are largely overlooked here. This gives the impression that the authors have not fully considered or integrated the existing body of research, weakening the paper's scientific foundation. The authors should narrow their focus to the specific implications of their results for Dome A. Any comparisons should be limited and carefully contextualized. A more detailed engagement with existing firn literature and a clearer articulation of the glaciological relevance of their findings would significantly improve the manuscript.

As it stands, I believe the paper would benefit from Major Reviews. While the Dome A data, and the proof that anthropogenic noise can be used to produce a model of firn, is a valuable contribution to the glaciological community, the current framing, discussion, and treatment of the literature do present itself in an appropriate way.

**Dear Anonymous Reviewer 2,**

Thank you very much for your constructive comments and suggestions! We have carefully revised our manuscript accordingly. We have focused the implications of our study on Dome A and highlighted the scientific glaciological significance of our finding about firn structure in the revised manuscript. Please find our point-to-point responses below.

The line numbers used in this response letter refers to the manuscript with marks.

**Introduction:**

1.The introduction would benefit from a clearer and more cohesive structure that better aligns with the specific focus of the study—imaging firn at Dome A. Currently, it opens with a broad discussion on Antarctica's vulnerability to climate change but does not make a direct connection to the relevance of firn studies within that context. For a journal like The Cryosphere, such general framing may not be necessary and could be replaced with a more targeted explanation of why firn structure and compaction at Dome A are scientifically important.

We followed your suggestion and reduced the general background description of Antarctica (Lines 33-39). We added information on the scientific significance of firn research, thereby making the introduction more focused on the shallow firn structure targeted in this study:

"Firn, formed through snow accumulation and subsequent compaction, represents the transitional layer between snow and glacial ice. It is an essential component of the ice sheet and plays a crucial role in material transport (MacAyeal, 2018). Its structure and evolution are influenced by processes such as densification, settling, and refreezing, which are highly sensitive to temperature variation, surface accumulation, and wind patterns (Ligtenberg et al., 2011; Wilkinson, 1988). Understanding firn dynamics is essential for accurately assessing surface mass balance, especially in Antarctica and Greenland (Gardner et al., 2018; Kowalewski et al., 2021; Velicogna et al., 2020). Moreover, firn modulates the depth at which atmospheric gases are sealed into the ice, directly impacting the interpretation of ice core records and paleoclimate reconstructions (Schwander et al., 1997). Variations in firn density and related physical properties affect the retrieval and interpretation of ice sheet elevation changes (Medley et al., 2022; Smith et al., 2023). Firn layers can store meltwater seasonally in the form of firn aquifers, influencing subglacial hydrology and potentially enhancing basal sliding (Forster et al., 2014; Miller et al., 2018). These multifaceted roles establish firn as a critical component in both observational and modeling efforts aimed at improving our understanding of polar ice sheet evolution and mass changes." (Lines 39-51 in the revised manuscript)

Dome A is the highest point of the East Antarctic Ice Sheet and differs significantly from West Antarctica and other coastal regions. It allows for longer preservation times of ice layers and more intact ancient ice records. Moreover, it offers a unique setting where characterizing the subsurface firn structure provides valuable insights into ice dynamics across the East Antarctic Plateau. In the revised manuscript, we have further added descriptions to highlight the significance of studying firn structure at Dome A:

"Although the shallow firn layer is known to be important, detailed investigations on the East Antarctic Plateau are still limited. Prior studies have demonstrated that the mechanical properties of firn can be influenced by ice crystal anisotropy at depths down to 100 m (Schlegel et al., 2019; Gerber et al., 2023; Pearce et al., 2024). Thus, applying seismic ambient-noise studies at Dome A can provide new insights into firn structure across high-elevation regions of the East Antarctic Plateau." (Lines 75-79 in the revised manuscript)

2. Additionally, the paragraph on seismic methods in Antarctica includes discussion of active-source techniques, which are not directly relevant to a study using ambient noise. A more focused discussion on ambient noise methods would strengthen the context. It would also be helpful to reference relevant studies using ambient noise to investigate firn structure, even if conducted outside Antarctica, since the methodology and findings are directly comparable to the present study.

We have added relevant studies using ambient-noise methods to investigate firn/ice structures in the revised manuscript:

"Similar ambient-noise studies have also been performed in Greenland, Glacier d'Argentière (France), Gornergletscher (Switzerland), Aletschgletscher (Switzerland) and de la Plaine Morte (Switzerland) (e.g., Pearce et al., 2024; Sergeant et al., 2020; Preiswerk and Walter, 2018; van Ginkel et al., 2025) to investigate glacier structures. These studies provided important insights into the subsurface structure of firn and ice." (Lines 63-66 in the revised manuscript)

We followed your suggestion and reduced the description of active-source methods, grouping them with radar, gravity, and other techniques (Lines 61-63 in the revised manuscript).

**Data & Methods:**

3.This section could be improved by restructuring for clarity and cohesion. At present, it reads somewhat like a list of loosely connected points. For example, Line 95 begins with a description of data processing, followed immediately by a sentence about station A's location—two pieces of information that could be better integrated into a more logically flowing narrative.

We have revised the section to improve its clarity and logical coherence, including minor wording and phrasing adjustments in several sentences (Lines 95-123):

"Dome A is located approximately at the central point of Line 1." (sentence moved from Line 115 to Line 95 in the revised manuscript)

"We first cut the ambient-noise data into 10-minute segments. Then, we applied both running absolute mean normalization and spectral whitening to the data. Subsequently, we used cross-correlation and phase-weighted stacking (Schimmel and Paulssen, 1997) to recover empirical Green's functions." (Lines 117-119 in the revised manuscript)

4.Please clarify the process used to forward model the dispersion curves, specifically how different modes were identified and associated. Additionally, it is stated that density is derived from Vs, although $V_{SH}$ is used—this should be corrected or clarified.

(1) The forward problem is formulated as solving a nonlinear implicit function:
$$F(Vr, f; Vs, Vp, \rho, h) = 0,$$
where $Vr$ is the phase velocity to be solved, $f$ is frequency, and $Vs$, $Vp$ (no need for Love wave), density $\rho$ and layer thickness $h$ are from our layered model. We apply a bisection algorithm to numerically search for the roots at each frequency. Multiple modes are identified by incrementally tracking zero-crossings corresponding to higher-mode solutions.
We clarify this process in the revised manuscript (Lines 142-144):

"The forward simulation of dispersion curves is solved with a Knopoff's method (Knopoff, 1964). For each frequency, we can obtain a set of solutions that correspond to fundamental and higher modes sequentially."

(2) We have corrected $V_S$ with $V_{SH}$ in the revised manuscript. (Line 153)

5. Several statements such as "fairly well" appear throughout the manuscript; these should be quantified where possible to improve scientific rigor. For instance, when discussing the fit between the derived density profile and core data, numerical metrics or visual comparisons would be helpful.

We quantified the misfit by using the root-mean-square error (RMSE) and added these values in the revised manuscript (Table A1 and Table A2 in Appendix A).
We calculated the density results under different empirical relationships (SV velocity-density from Yang et al. (2024) and SH velocity-density from Diez et al. (2014)) and ice density values (900 and

920 kg m⁻³), and the corresponding RMSEs are summarized in Table A1. We also calculated the RMSEs of the multi-modal dispersion curves fitting (Table A2). The low RMSEs quantitatively support the reliability of the fitting results:

"The RMSEs between the density (mean and knee point results) and ice core data are less than 44 kg m⁻³ (Table A1 in Appendix A)." (Line 174 in the revised manuscript)

"The RMSEs between the synthetic and observed dispersion curves are less than 34 m s⁻¹ (Table A2 in Appendix A)." (Line 180 in the revised manuscript)

**Table A1**. Root-mean-square errors (RMSE) between estimated density results (mean and knee point results) and ice core (DA2005) density.

| Empirical Relationships | $\rho_{ice}$ (kg m$^{-3}$) | Mean RMSE (kg m$^{-3}$) | Knee point RMSE (kg m$^{-3}$) |
|---|---|---|---|
| Diez et al., 2014 (SH velocity-density) | 900 | 43.8465* | 43.5244* |
| Diez et al., 2014 (SH velocity-density) | 920 | 46.2571 | 44.8456 |
| Yang et al., 2024 (SV velocity-density) | 920 | 47.0541 | 42.3032 |

\* indicates the empirical relationship and ice density value used in the main text.

**Table A2**. Root-mean-square errors (RMSE) of multi-mode dispersion curve fitting.

| Dispersion Modes | Love Wave | | Rayleigh Wave | |
|---|---|---|---|---|
| | Mean RMSE (m s$^{-1}$) | Knee point RMSE (m s$^{-1}$) | Mean RMSE (m s$^{-1}$) | Knee point RMSE (m s$^{-1}$) |
| Mode 0 | 22.3250 | 17.0714 | 33.9394 | 25.7128 |
| Mode 1 | 28.0159 | 28.1090 | 22.5358 | 29.8653 |
| Mode 2 | 11.9490 | 8.3569 | 18.2726 | 19.9725 |
| Mode 3 | 14.0678* | 15.2696* | 14.5797* | 14.3996* |
| Mode 4 | — | — | 21.4708* | 28.0765* |

\* indicates the mode not involved in the dispersion curve inversion.

6. On Line 165, the interpretation that faster $V_{SV}$ than $V_{SH}$ in the shallow firn is due to vertically aligned snow grains needs further explanation. Please elaborate on the physical mechanism behind this interpretation, and consider whether snow grain settling alone can produce this effect. It would also be beneficial to include a sensitivity analysis to demonstrate which depth ranges are constrained by the frequencies used in the inversion. Providing separate plots of $V_{SH}$ and $V_{SV}$, in addition to their ratio, in the supplementary material would also help readers interpret the results more fully.

(1) We thank the reviewer for the valuable suggestion. We have removed the interpretation of shallow weak anisotropy (< 5 %) in the revised manuscript (Line 192). Because our observations lack data at shorter wavelength ranges (< 40 m), and the anisotropy results in the shallow layer (< 20 m) exhibit high uncertainty (Fig. 4a).

We have added explanations of the anisotropy in the revised manuscript (Lines 212-232):

"Previous studies (e.g., Pearce et al., 2024; Chaput et al., 2023; Diez et al.,2016; Picotti et al., 2015; Schlegel et al., 2019) show that firn anisotropy may be caused by three primary mechanisms: (1) effective anisotropy, related to very thin layers formed during firn densification; (2) structural anisotropy, related to fractures or microcracks caused by non-isotropic stress; and (3) intrinsic anisotropy, associated with preferred crystal orientation.

In the shallow layer (< 20 m), some lateral variation may exist along Line 1 (~10 km). Our observations lack data at shorter wavelength ranges (< 40 m), and the weak anisotropy results (< 5 %) in the shallow layer exhibit high uncertainty (Fig. 4a). Therefore, we refrain from further interpretation of the anisotropy within the upper 20 m.

The mean accumulation rate in the Dome A region is about 2.3 cm water equivalent per year (Jiang et al., 2012) and, therefore, firn densification can produce millimeter-scale layers whose thicknesses are much smaller than the seismic wavelengths observed in our study. These thin layers lead to different elastic properties in the vertical and horizontal directions (Diez et al.,2016; Schlegel et al., 2019). It makes the SH waves travel relatively faster than SV waves along the horizontal direction, which is consistent with our observation.

The average strain rate in the Dome A region ($\sim$1.6$\times$10$^{-12}$ s$^{-1}$) (Yang et al., 2014) is much lower than the typical ductile to brittle transition values for firn ($\sim$10$^{-4}$ to 10$^{-2}$ s$^{-1}$) (Narita, 1984; Kirchner et al., 2001), suggesting that tensile stresses are unlikely to cause brittle crevasse in our study area. Therefore, the structural anisotropy caused by crevasse is not considered the main reason for anisotropy in the Dome A region.

Regarding intrinsic anisotropy, the preferred orientation direction of ice crystals is related to gravitational compaction and ice flow. For the firn layer after reaching the first critical depth (28.1 m), slow ice flow ($\sim$11.1 cm a$^{-1}$) that is perpendicular to the Line 1 (Yang et al., 2014) causes re-orientation of the crystal along the horizontal direction. This leads to SH waves traveling faster than SV waves. Overall, we infer that the observed anisotropy at Dome A region is primarily attributed to effective anisotropy and intrinsic anisotropy."

(2) We showed the sensitivity kernels of multi-modal Love and Rayleigh waves (Fig. A2, which is also added in Appendix A in the revised manuscript). Overall, the inclusion of higher modes extends the usable frequency range, enhances the constraints on the inversion, and improves the accuracy of the results.

[Figure]

**Figure A2**. Depth sensitivity kernels at different frequencies for Rayleigh (a-c) and Love (d-f) waves for the fundamental (mode 0), first high (mode 1), and second high (mode 2) mode. The black dotted lines represent the dispersion curves for each mode.

(3) In the original manuscript, Figure 3a and b show the solutions of $V_{SH}$ and $V_{SV}$, respectively. The radial anisotropy ($V_{SH}/V_{SV} - 1$) shown by the grey lines in Figure 4a is calculated from all solutions in these two sets. To help readers better understand the relationship between $V_{SH}$ and $V_{SV}$, we have now plotted their solution sets and mean solutions in a single figure (Fig. A3, which is also added in Appendix A in the revised manuscript) using different colors for clarity.

[Figure]

**Figure A3**. Pareto solution sets of $V_{SH}$ (light red lines) and $V_{SV}$ (light blue lines). The mean solutions of $V_{SH}$ and $V_{SV}$ are marked by red and blue lines, respectively. These results are the same as those presented in Figure 3a and b of the main text.

**Discussion:**

7.The discussion would benefit from a sharper focus on the specific insights gained from studying firn at Dome A. While comparing firn conditions across East and West Antarctica may seem appealing, such comparisons must be made cautiously due to the highly localized nature of firn compaction processes. The current manuscript draws broad conclusions about regional differences that are not strongly supported by the data and may oversimplify the complexity of firn behaviour. To strengthen the paper, the discussion should centre on the local significance of the Dome A firn profile, and what new insights are gained from applying ambient noise methods in this specific setting.

We accepted your suggestion and revised the discussion accordingly. In the revised manuscript, we have emphasized that our results are based on one-dimensional velocity structures with limited spatial coverage, and thus may only reflect local differences in the regions where data were collected. These constraints are not sufficient to comprehensively assess the differences between East and West Antarctica:

[revised manuscript text omitted]

---

## Author Comment (AC3)

**Response to Anonymous Reviewer 3**

This paper presents a welcome iteration in ambient noise methods applied to imaging firn media, with a focus here on Dome A in East Antarctica. The authors use data collected from linear nodal arrays and leverage human camp noise to recover multi-modal surface waves to perform a near-surface velocity inversion. The paper is methodologically simple, in that the methods are well established, and the results are well presented.

As do the other reviewers, I have a few issues with the interpretation of results and the relative simplicity of the analysis in the global context of firn formation and structure:

**Dear Anonymous Reviewer 3,**

Thank you very much for your constructive comments and suggestions! We have carefully revised our manuscript accordingly. Please find our point-to-point responses below.

The line number used in this response letter refers to the manuscript with marks.

1. Fig 3c: I agree with reviewer 1 on the velocity/density relationship you used here from Diez 2014, and the probable necessity for an updated form. The accumulation and strain environment of West Antarctica significantly differs from East Antarctica, and that could easily account for your disparities.

Thank you for your suggestion and we compared the results obtained using different parameter values in the empirical relationship.

(1) The results using different ice densities (900 and 920 kg m⁻³, Figure A1a) are very similar. Relatively speaking, the estimated density profile using an ice density of 920 kg/m³ is slightly closer to the borehole data (DA2005) in the region shallower than 50 m depth, and slightly further from DA2005 in the region deeper than 50 m depth (blue and yellow lines in Figure A1a). It indicates that the choice of ice density has a minor impact on the result.

(2) In Figure A1b, the density result also generally agrees with the borehole data (DA2005). The result based on the relationship from Yang, Zhan et al. (2024) improves the fit within the upper ~30 m compared to that of Diez et al. (2014) but also shows some deviations at greater depths. The RMSE results (Table A1) indicate that the densities estimated using different empirical relationships and ice density values generally agree with the ice core data.

(3) We also agree that a site-specific velocity-density relationship may be beneficial. Given the spatial distance and difference in acquisition time between the borehole and our linear array, the borehole data serve only as a reference density result. Further borehole measurements, ideally closer in both space and time to the array, may help to establish a representative empirical relationship near the Dome A region.

We added this information in the Appendix A (Figure A1 and Table A1).

[Figure]

**Figure A1**. Density estimated by empirical relationship. **(a)** Using a relationship from Diez et al. (2014) with $V_{SH}$. The red and blue lines use an ice density of 900 kg m$^{-3}$, while the green and yellow lines use 920 kg m$^{-3}$. **(b)** Using the relationship from Yang, Zhan et al. (2024) with $V_{SV}$ (grey, green and yellow lines). The filled circles show the ice core (DA2005) density near Dome A. The red and blue lines in (a) and (b) are consistent with Fig. 3c in the main text.

Table A1. Root-mean-square errors (RMSE) between estimated density results (mean and knee point results) and ice core (DA2005) density.

| Empirical Relationships | $\rho_{ice}$ (kg m$^{-3}$) | Mean RMSE (kg m$^{-3}$) | Knee point RMSE (kg m$^{-3}$) |
|---|---|---|---|
| Diez et al., 2014 (SH velocity-density) | 900 | 43.8465* | 43.5244* |
| Diez et al., 2014 (SH velocity-density) | 920 | 46.2571 | 44.8456 |
| Yang et al., 2024 (SV velocity-density) | 920 | 47.0541 | 42.3032 |

* indicates the empirical relationship and ice density value used in the main text.

2. Firn anisotropy in East Antarctica very likely can be explained almost entirely from radial anisotropy given the minimal ice flow, but when comparing to West Antarctica, you should be cautious. For instance, flowing ice and firn has been shown to have multiple azimuthal anisotropy mechanisms related to fracturing and firn plasticity, and these can impact estimates of the magnitude of radial anisotropy estimates. Have a look at:

Chaput J, Aster R, Karplus M, Nakata N, Gerstoft P, Bromirski PD, Nyblade A, Stephen RA, Wiens DA (2023). Near-surface seismic anisotropy in Antarctic glacial snow and ice revealed by high-frequency ambient noise. Journal of Glaciology 69(276), 773 – 789. https://doi.org/10.1017/jog.2022.98

Advected fractures and other embedded features can also affect the local velocity model in West Antarctica firn, so it's certainly worth mentioning the ultra-local side of firn profiles.

Expanding on the discussion in terms of firn formation differences would be an improvement here, though given that edits will be largely constrained to discussion (with the exception of perhaps a test related to point 1 above), I think minor reviews are appropriate.

(1) Thank you for your valuable suggestion. We have added explanations of the anisotropy results in the revised manuscript (Lines 212-232):

[revised manuscript text omitted]

---

## Author Response (AR2)

**Response to Anonymous Reviewer 2**

Thank you for the corrections you have made to your manuscript. I found the responses to my review very through and believe that the updated manuscript has benefited massively from the changes made. The paper now feels very targeted towards a cryo-seismology and glaciological community who will be able to directly benefit from the method and results published. However, I find the discussion of the paper is still lacking in critically assessing the results that they have published. Therefore, I am happy for this paper to be published assuming the following is addressed.

Thank you very much for your constructive comments and suggestions! We have carefully revised our manuscript accordingly. Please find our point-to-point responses below. The line numbers used in this response letter refer to the manuscript with marks.

**Discussion:**

1. The author has removed the majority of the text comparison between East and West Antarctic firn, but has kept the figures in comparing results. They have not added any additional discussion on what the firn profile from Dome A actually shows, why it is important for that region or what the results tell us. They do reference what this means for the grain structure of Dome A in the section above, which perhaps could be moved to the discussion.

We described the meaning of the Dome A study in the "Introduction" (Lines 63-72). We also added descriptions about this study in Dome A in the "Discussion" in the revised manuscript as:

"Most of the previous studies in Antarctica report density with either  $V_{SH}$  or  $V_{SV}$ , we showed that joint analysis of  $V_{SH}$ ,  $V_{SV}$ , and density provides a more comprehensive characterization of the shallow structure at Dome A. It improves our understanding of the transition of snow, firn and glacier ice at this high point in East Antarctica." (Lines 225-228 in the revised manuscript)

2. The author also acknowledges that they could assess the variation in firn along/across profile but do not do this. This is a shame, and would have been a valuable addition to the paper and formed a relevant discussion on what the results mean for Dome A.

Thank you for your suggestion. In our study, Line 1 is deployed right across Dome A, while the central point of Line 2 is more than 10 km away from Dome A. Our study focuses on the shallow firn structure (~100 m depth) at Dome A. Therefore, we didn't assess azimuthal anisotropy, which represents an averaged value at Kunlun station rather than Dome A.

Figure R1 (Figure 1a in the manuscript). Arrays information of Line 1 (black triangles), Line 2 (black hollow triangles), and Line 3 (white triangles). The Kunlun Station (red pentagram), Dome A and the ice core (DA2005) (red circle) are also marked.

3. Currently, the discussion feels very disconnected from the rest of the paper, and a bit irrelevant to the findings. Again, comparing the results from East/West Antarctica does not feel relevant, and the way the author has referenced the results does not pose itself to be a strong enough discussion to justify why the comparison is being made. I believe, if you are going to compare results from different stations/parts of Antarctica, then a comparison between the firn profiles individually should be made, rather than grouping the models into East/West (e.g. perhaps a comment on how the firn profile is thinner/thicker at Dome A than what is expected for the given temperatures/accumulation rate?). I believe you can keep in your comparison of individual sites in antarctica but referencing a sweeping statement of East/west locations is not a relevant comparison.

We followed your suggestion and removed the broad comparison between East and West Antarctica. We focus our discussion on the specific test sites where the referenced Vs were obtained in the revised manuscript. We also removed the East and West Antarctica classification and added the specific regions in Figure 5a.

"Furthermore, we compared several existing S-wave velocity profiles of firn structures from different areas in Antarctica, which indicate relatively higher S-wave velocities at the same depth in the study areas located in West Antarctica." (Line 27-29 in the revised manuscript)

"Figure 5a suggests relatively higher  $V_S$  at the same depth in the shallow regions in the Ross Ice Shelf, the Rutford Ice Stream, the Whillans Ice Stream, and the WAIS Divide camp. The higher  $V_S$  may indicate thinner snow layers and a more rapid transition from snow to firn and ice in these areas." (Line 229-231 in the revised manuscript)

Figure R2 (Figure 5a in the manuscript). S-wave velocity from different areas in Antarctica. (a)  $V_{SH}$ ,  $V_{SV}$  from this study (blue line with pentagrams and blue line),  $V_{SV}$  from South Pole (Yang et al., 2024; green curve with circles),  $V_{SH}$  and  $V_{SV}$  from Ross Ice Shelf (Diez et al., 2016; pink curve with triangles and pink dotted curve with triangles),  $V_{SV}$  from Rutford Ice Stream (Zhou et al., 2022; yellow dotted curve with diamonds),  $V_{SH}$  from Whillans Ice Stream (Picotti et al., 2024; red dash-dot curve with inverted triangles),  $V_{SV}$  and  $V_{SV}$  from WAIS Divide camp (Qin et al., 2024, Zhang et al., 2024; brown dashed curve with squares and purple line with squares).

4. Perhaps what would be a better comparison is to compare the ratio of Vsv to Vsh for each site, so the comparison between firn models can be made irrespective of location. To show how the vsv/vsh ratio is repeatable (or not?) irrespective of location would be a useful discussion.

Most previous studies report either  $V_{SH}$  or  $V_{SV}$  alone (please refer to Figure 5a, which is shown in our response to point 3). Therefore, we are unable to estimate the  $V_{SV}/V_{SH}$  ratio for most sites.

5. Line 90: 'We deployed a total of 73 three component seismic nodes' – I do not understand why you do not use all 73 stations in your results. Please add a line to explain why you only use the 19 stations.

In this manuscript, we mainly used data from Line 1 to estimate the radial anisotropy in the Dome A region, as the other stations are located farther from Dome A compared to Line 1. We added an explanation in the revised manuscript (Lines 81-82):

"In this study, we mainly used 19 seismic nodes with a 500 m interval (Line 1, which was deployed across Dome A), to reconstruct the shallow structure at Dome A region."